# OpenReview forum: "Modulating LLM Behavior via Context-Specific Activation Steering"
_ICLR.cc/2026/Conference — Submitted to ICLR 2026_

### Official Review · Reviewer_Mobr · 2025-10-23

**Soundness:** 2
**Presentation:** 2
**Contribution:** 2
**Rating:** 4
**Confidence:** 4

**Summary:**

This paper proposes Context-Specific Steering (COS-Steering), which replaces static, hand-crafted activation categories with dynamic steering. The method compresses a pool of steering signals into basis vectors via sparse autoencoders and learns input-specific weights for context-aware control. Experiments show improved robustness to adversarial prompts while maintaining low impact on benign queries.

**Strengths:**

- The paper introduces a context-specific steering mechanism that adapts dynamically to each input, addressing limitations of static, hand-crafted activation categories.
- The proposed method demonstrates strong effectiveness in alignment tasks, showing improved robustness against adversarial prompts.

**Weaknesses:**

**Major concern:**

The claim (Line 12) that “current activation engineering methods embed a static premise” is not entirely accurate, as important related work is missing. In particular, PaCE (NeurIPS 2024) introduces activation steering on-the-fly via sparse coding, which directly challenges this claim.
- It would be helpful to clarify what specific challenges remain unresolved by PaCE that your method addresses.
- In addition, please elaborate on why your proposed approach remains necessary in light of PaCE.
- Strengthening the motivation section by positioning your contribution more clearly relative to PaCE would further highlight the novelty and significance of your work.

*Ref: Luo, J., Ding, T., Chan, K. H. R., Thaker, D., Chattopadhyay, A., Callison-Burch, C., & Vidal, R. (2024). Pace: Parsimonious concept engineering for large language models. Advances in Neural Information Processing Systems, 37, 99347-99381.*

**Other concerns:**
- Line 196: Could you clarify the criterion used to divide the bad cases dataset?
- Line 251: The Orthogonal Representation Penalty $P_{OR}$ appears to be constant with respect to the optimization variable $S_{r_l}$​. Could you please elaborate on how this term influences the optimization process?
- In Table 1, evaluation on benign prompts is missing. In particular, it would be valuable to report metrics such as BERTScore and KL divergence when applying your method to benign inputs. This would help assess whether the proposed approach introduces unintended distortions or shifts in non-adversarial settings.
- As the experiments are limited to Llama-2 and Vicuna, I am wondering how the method would perform on more recent LLMs such as Llama-3-8B. Including results or discussion on newer models would strengthen the empirical evaluation and provide a clearer picture of the method’s generality.
- Most experiments do not report statistical uncertainty. Adding standard deviations or confidence intervals would make the results more reliable.

**Minor comment:**
- Line 425: The link to the Appendix appears to be broken.

**Questions:**

Please see **Weaknesses** for my questions.

---

> ### Author Response · Authors · 2025-11-18
> **Rebuttal for Reviewer Mobr 1/2**
>
> ### **Rebuttal for Reviewer Mobr**
>
> We sincerely thank you for insightful feedback and constructive suggestions. These comments are invaluable for enhancing the clarity, rigor, and impact of our work. We have carefully considered each point and will address them below.
>
> ---
>
> ### **Response to Major Concern:**
>
> > **Reviewer's Concern:** The claim (Line 12) that "current activation engineering methods embed a static premise" is not entirely accurate, as important related work is missing. In particular, PaCE (NeurIPS 2024) introduces activation steering on-the-fly via sparse coding, which directly challenges this claim.
>
> We thank reviewer Mobr for this highly inspiring work that steers the model by zeroing-out its harmful-semantic component during inference. And this paper has been added to the Related Work Appendix B.3 (marked as blue). Yet PaCE does not resolve the static-premise issue; it merely obeys that premise in a different way. To be specific, PaCE **suffers** from the core flaw of the static premise: **over-reliance on hand-crafted semantic categories**. We will clarify our paper's position from two aspects: the paper’s intended scope and other related considerations.
>
> **Scope: PaCE also Shares the Root Cause of The Static Premise**
>
> The static-premise arises from an over-reliance on hand-crafted semantic categories. Although PaCE lets GPT automatically label the single "harmful" component, **the labeling rule still follows the hand-crafted harm categories**, risking the omission of subtle harmful semantics embedded in diverse attack methods.
>
> For example, CAST(ICLR 2025 Spotlight)[1] performs conditional activation steering that depends on the input, yet collapses under adversarial prompt variations because it is still anchored to hand-crafted semantic categories. PaCE departs from CAST in that its pre-defined category is monosemantic; the harmfulness label itself, however, is still obtained from annotation rules that follow hand-crafted semantic categories. **Both approaches inspect the input prompt for labels confined to hand-crafted categories before intervening, rather than generating a context-specific steering vector on-the-fly.**
>
> In contrast, COS-Steering eliminates dependence on hand-crafted semantic categories by adaptively constructing a safety-steering subspace, instantiates a context-specific steering vector within the subspace.
>
> **Other Considerations**
>
> **Cost**. COS-Steering incurs lower training cost than PaCE because it avoids large-scale unsupervised pre-training. Moreover, its parameters can be fused into the base model, eliminating any inference overhead, whereas PaCE requires a larger set of additional parameters that remain separate and impose extra cost per query.
>
> **Use of SAE**. In PaCE, the SAE is used to extract a single semantic component. This is the standard application of SAEs with LLMs. In COS-Steering, however, the SAE functions only as a compressor that yields steering vectors. These two goals are fundamentally different.
>
>
> Nevertheless, your valuable feedback has alerted us to the necessity of clarifying this issue; we have accordingly added an explicit discussion in the revised paper.
>
> [1] Bruce W. Lee, Inkit Padhi, Karthikeyan Natesan Ramamurthy, Erik Miehling, Pierre L. Dognin, Manish Nagireddy, Amit Dhurandhar: Programming Refusal with Conditional Activation Steering. ICLR 2025

---

> ### Author Response · Authors · 2025-11-18
> **Rebuttal for Reviewer Mobr 2/2**
>
> ### **Response to Other Concern:**
>
> > **Reviewer's Concern:** Line 196: Could you clarify the criterion used to divide the bad cases dataset?
>
> Each group contains only 5-10 randomly sampled bad cases; optimizing a steering vector on such a small set is straightforward, so under-fitting is not a concern. In this setting we view the random sampling as a simple form of regularization that further lowers any risk of over-fitting.
>
> Perplexity and over-refusal rates have been incorporated into the evaluation and are reported in Table 1.
>
> > **Reviewer's Concern:** Line 251: The Orthogonal Representation Penalty  $P_{OR}$    appears to be constant with respect to the optimization variable Sr. Could you please elaborate on how this term influences the optimization process?
>
> $Sr$ is trained on both harmful and harmless inputs.  $P_{OR}$    is applied only to harmless inputs, where it is computed dynamically; for harmful examples no  $P_{OR}$    is needed, so the term is fixed at the constant 0.
>
> > **Reviewer's Concern:** In Table 1, evaluation on benign prompts is missing. In particular, it would be valuable to report metrics such as BERTScore and KL divergence when applying your method to benign inputs. This would help assess whether the proposed approach introduces unintended distortions or shifts in non-adversarial settings.
>
> As stated on Section 4.1 Metrics, both KL divergence and BERTScore were already computed exclusively on harmless prompts, exactly as you suggested.
>
> Additionally, other reviewers requested direct evaluation of generated-text quality and the over-refusal rate; we consolidate those results here for your convenience.
>
> | Type               | Method         | Advbench PPL↓ | Advbench ORR↓ | HarmfulQA PPL↓ | HarmfulQA ORR↓ |
> |--------------------|----------------|---------------|---------------|----------------|----------------|
> | Decoding Time      | Self-Reminder  | 15.49         | **0.01**      | 5.84           | **0.01**       |
> | Decoding Time      | SafeDecoding   | 7.18          | **0.02**      | **1.07**       | 0.04           |
> | Activation Eng.    | MeanCentring   | **7.14**      | 0.21          | 25.25          | 0.09           |
> | Activation Eng.    | BiPO           | 45.49         | 0.05          | 24.17          | **0.01**       |
> | Activation Eng.    | ITI            | 13.70         | **0.01**      | **4.81**       | 0.20           |
> | Activation Eng.    | CAST           | 32.67         | **0.02**      | 32.67          | **0.02**       |
> | **Ours**           |**COS-Steering**| **3.83**      | **0.02**      | 7.99           | **0.01**       |
>
> **Evaluation of generation quality.** Table 1 uses KL divergence and BERTScore to quantify the deviation from the original model's distribution after alignment. As a complementary metric, perplexity is adopted to directly assess generation quality on benign prompts, shown in Table above. The results show that COS-Steering keeps perplexity at a low level overall. To avoid evaluation circularity, the scores are computed with Qwen3-8B.
>
> **Over-refusal rate.** Over-refusal rate is a key metric for gauging the excessive sensitivity of a LLM's safety mechanism; it measures the proportion of benign prompts that the model incorrectly refuses to answer. Table above shows that COS-Steering maintains a low over-refusal rate.
>
> > **Reviewer's Concern:** As the experiments are limited to Llama-2 and Vicuna, I am wondering how the method would perform on more recent LLMs such as Llama-3-8B. Including results or discussion on newer models would strengthen the empirical evaluation and provide a clearer picture of the method's generality.
>
> Experiments are conducted on Llama-3-8B; results are reported below.
>
> | Method  | Advbench | HarmfulQA|
> |:-------:|:--------:|:-------:|
> |Llama3-8B|  0.10    |   0.05  |
> |+COS-Steering|  0.01  |  0.01 |
>
> > **Reviewer's Concern:** Most experiments do not report statistical uncertainty. Adding standard deviations or confidence intervals would make the results more reliable.
>
> We will update this part in the forthcoming revision.
>
> > **Reviewer's Concern:** Line 425: The link to the Appendix appears to be broken.
>
> The broken reference on have been corrected.
>
> ---
>
> We are very grateful for this detailed list of suggestions, which will significantly improve the paper's quality and readability. If any concerns remain, or if new questions arise, we would be grateful to hear from you.

---

> ### Author Response · Authors · 2025-11-28
> **Report of Statistical Uncertainty**
>
> > **Reviewer's Concern:** Most experiments do not report statistical uncertainty. Adding standard deviations or confidence intervals would make the results more reliable.
>
> To enhance reliability, Table 2 now reports statistical uncertainty.
>
> We are very grateful for this detailed list of suggestions, which will significantly improve the paper's quality and readability. We sincerely would like to know whether our response has adressed your concerns. If any concerns remain, or if new questions arise, we would be grateful to hear from you.

---

> ### Comment · Area_Chair_qDtH · 2025-11-28
> **A gentle reminder to participate in the author–reviewer discussion.**
>
> Dear Reviewer Mobr,
>
> Thank you once again for your service to ICLR 2026. Now that the authors have submitted their rebuttal, could you please engage in the interactive discussion with them? Your participation would be very helpful to the authors, and they would greatly appreciate it. Please also read the authors’ response together with the other reviews and consider whether the rebuttal or any additional comments influence your assessment of the paper.
>
> Thank you again for your efforts.
>
> Best wishes,
>
> Your AC

---

### Official Review · Reviewer_kKNM · 2025-10-24

**Soundness:** 2
**Presentation:** 2
**Contribution:** 3
**Rating:** 2
**Confidence:** 3

**Summary:**

The paper studies how to guide large language models (LLMs) to refuse harmful prompts without blocking benign ones. Earlier activation‑steering techniques assume that each input can be mapped into a fixed semantic category and apply a single pre‑computed direction to the hidden activations of the model. The authors observe that adversarial jailbreak prompts can shift activations away from these hand‑crafted categories, leading to missed detections or excessive refusals. They propose **Context‑Specific Steering (COS‑Steering)**. The approach collects many safety‑related steering vectors and compresses them into a sparse basis using a sparse autoencoder. At inference time, a small controller reads the current input’s activation and outputs coefficients over the basis, producing a steering vector tailored to that input. The authors test the method on a mixed‑attack scenario that combines multiple jailbreak attacks. COS‑Steering maintains high refusal rates on harmful prompts and introduces little degradation on harmless queries compared with baseline steering methods.

**Strengths:**

- **Originality**. COS‑Steering replaces fixed, category‑based steering with a dynamic mechanism that weights multiple safety directions according to the input. This addresses a known limitation of activation steering and provides a new way to integrate sparse features into steering.

- **Method design**. Using a sparse autoencoder to derive basis vectors captures diverse steering signals while keeping the controller small. The lightweight module that reads activations and outputs weights allows context‑dependent interventions with minimal overhead.

**Weaknesses:**

-   **Missing LLM usage and reproducibility statements**. These are strongly recommended for ICLR submissions. Missing these key statements makes it difficult to evaluate the reliability of the work. While there are novel contributions in this paper, the authors should at least clarify details for creating the work before I raise my score.
-   **KL divergence is not sufficient to validate consistent output quality**. In the experiments, ASR and Harmful Score are used to evaluate safety, while BERTScore and KL divergence are used to measure performance retention. However, BERTScore and KL divergence are not very robust metrics for evaluating output changes and are insensitive to generation quality or over-refusals. The paper could report results on perplexity changes and over-refusal rates to further confirm the selectivity for safety responses.
-   **Lack of clear evidence on Sr choosing decoder vectors selectively**. While the paper's core motivation is that Sr will activate Sd vectors based on context, we need direct evidence of this. The negligible side effects on benign queries could be explained by the fact that the paper only trains a small adapter on the model to refuse harmful queries, similar to LoRA.
-   **There are some syntax errors in the paper**. For example, a wrong quote mark direction in Line 199 and a missing reference (Appendix ??) in Line 425.

**Questions:**

-   In Line 413: how is the "top-10% activation range" defined?
-   For Figure 5, why are there multiple bars for one layer? Are they different heads in that layer?
-   What if we don't train Sr and Sd separately, but train the whole adapter in one training session? What is the fundamental difference between COS-Steering and a LoRA adapter?

---

> ### Author Response · Authors · 2025-11-18
> **Rebuttal for Reviewer kKNM 1/3**
>
> ### **Rebuttal for Reviewer kKNM**
>
> We sincerely thank you for insightful feedback and constructive suggestions. These comments are invaluable for enhancing the clarity, rigor, and impact of our work. We have carefully considered each point and will address them below.
>
> ---
>
> ### **Response to Weakness1: LLM usage and reproducibility statements**
>
> > **Reviewer's Concern:** Missing LLM usage and reproducibility statements. These are strongly recommended for ICLR submissions. Missing these key statements makes it difficult to evaluate the reliability of the work. While there are novel contributions in this paper, the authors should at least clarify details for creating the work before I raise my score.
>
> We apologize for missing these statements. LLMs were used solely for sentence-level polishing-i.e., re-phrasing, grammar checking, and disambiguating diction. A anonymized repository https://anonymous.4open.science/r/COS-Steering-2BC0 containing the complete pipeline has been submitted alongside this paper for reproducibility review. The complete code and data will be released after acceptance. And LLM usage and reproducibility statements have been added in Section 6 and is highlighted in blue.
>
> ---
>
> ### **Response to Weakness2: Evaluation-metric Adjustment**
>
> > **Reviewer's Concern:** KL divergence is not sufficient to validate consistent output quality. In the experiments, ASR and Harmful Score are used to evaluate safety, while BERTScore and KL divergence are used to measure performance retention. However, BERTScore and KL divergence are not very robust metrics for evaluating output changes and are insensitive to generation quality or over-refusals. The paper could report results on perplexity changes and over-refusal rates to further confirm the selectivity for safety responses.
>
> Using KL divergence to quantify the shift in the model's output distribution by comparing logits before and after alignment is an established practice in the literature. To comprehensively evaluation, perplexity and over-refusal rates have been incorporated into the evaluation and are reported in the table below and Table 4 in paper.
>
> | Type               | Method         | Advbench PPL↓ | Advbench ORR↓ | HarmfulQA PPL↓ | HarmfulQA ORR↓ |
> |--------------------|----------------|---------------|---------------|----------------|----------------|
> | Decoding Time      | Self-Reminder  | 15.49         | **0.01**      | 5.84           | **0.01**       |
> | Decoding Time      | SafeDecoding   | 7.18          | **0.02**      | **1.07**       | 0.04           |
> | Activation Eng.    | MeanCentring   | **7.14**      | 0.21          | 25.25          | 0.09           |
> | Activation Eng.    | BiPO           | 45.49         | 0.05          | 24.17          | **0.01**       |
> | Activation Eng.    | ITI            | 13.70         | **0.01**      | **4.81**       | 0.20           |
> | Activation Eng.    | CAST           | 32.67         | **0.02**      | 32.67          | **0.02**       |
> | **Ours**           |**COS-Steering**| **3.83**      | **0.02**      | 7.99           | **0.01**       |
>
> **Evaluation of generation quality.** Table 1 uses KL divergence and BERTScore to quantify the deviation from the original model's distribution after alignment. As a complementary metric, perplexity is adopted to directly assess generation quality on benign prompts, shown in Table above. The results show that COS-Steering keeps perplexity at a low level overall. To avoid evaluation circularity, the scores are computed with Qwen3-8B.
>
> **Over-refusal rate.** Over-refusal rate is a key metric for gauging the excessive sensitivity of a LLM's safety mechanism; it measures the proportion of benign prompts that the model incorrectly refuses to answer. Table above shows that COS-Steering maintains a low over-refusal rate.

---

> ### Author Response · Authors · 2025-11-18
> **Rebuttal for Reviewer kKNM 2/3**
>
> ### **Response to W3: Evidence on $Sr$ Choosing Decoder Vectors Selectively**
>
> > **Reviewer's Concern:** Lack of clear evidence on $Sr$ choosing decoder vectors selectively. While the paper's core motivation is that $Sr$ will activate $Sd$ vectors based on context, we need direct evidence of this. The negligible side effects on benign queries could be explained by the fact that the paper only trains a small adapter on the model to refuse harmful queries, similar to LoRA.
>
> We thank the reviewer for this critical question and fully agree on its centrality to the paper's rigor. Pivotal evidence is therefore provided from two perspectives-excluding benign scenarios and addressing different harmful categories.
>
> **Excluding Benign Scenarios**
>
> To directly contrast harmful and benign contexts, Figure 7 in Section 4.7 visualizes the activation weights that $Sr$ assigns to 16 SAE feature vectors across three layers. The plot shows that $Sr$ maintains low weights for all vectors in every benign instance, whereas in harmful prompts it selectively up-weights specific vectors to counter the threat.
>
> **Addressing Different Harmful Categories**
>
> To examine how feature vectors respond to diverse attacks, we inspect the category distribution of inputs whose weights rank in the top 10 % for each vector (Figure 5). Each bar depicts the distribution of high-weight inputs for one vector in $Sd$ of the corresponding layer. Several vectors are activated almost exclusively by a single attack method. It indicates that safety-steering vectors targeting individual attack methods have been encoded in $Sd$, and $Sr$ has learned to selectively activate these vectors according to the hidden states containing context.
> The 10 % threshold merely filters high-weight inputs; varying it (5 % or 1 %, provided in Appendix C.3) does not alter the conclusion.
>
> Nevertheless, your valuable feedback has alerted us to the necessity of clarifying this issue; we have accordingly added the Section 4.7 to discuss this issue in the revised paper.
>
> ---
>
> ### **Response to W4: Syntax Errors**
>
> > **Reviewer's Concern:** There are some syntax errors in the paper. For example, a wrong quote mark direction in Line 199 and a missing reference (Appendix ??) in Line 425.
>
> Both issues-the incorrect quotation-mark orientation on Line 199 and the missing reference on Line 425-have been corrected.

---

> ### Author Response · Authors · 2025-11-18
> **Rebuttal for Reviewer kKNM 3/3**
>
> ### **Response to Q1: Activation Range**
>
> > **Reviewer's Concern:** In Line 413: how is the "top-10% activation range" defined?
>
> The 10 % threshold merely filters high-weight inputs; varying it (5 % or 1 %, provided in Appendix C.3) does not alter the conclusion.
>
> ---
>
> ### **Response to Q2: Explaination of Figure 5**
>
> > **Reviewer's Concern:** For Figure 5, why are there multiple bars for one layer? Are they different heads in that layer?
>
> Each bar depicts the distribution of high-weight inputs for one vector in $Sd$ of the corresponding layer.
>
> ---
>
> ### **Response to Q3: Fundamental Difference between COS-Steering and LoRA Adapter**
>
> > **Reviewer's Concern:** What if we don't train $Sr$ and $Sd$ separately, but train the whole adapter in one training session? What is the fundamental difference between COS-Steering and a LoRA adapter?
>
> The comprehensive set of safety-steering features is fundamental difference between COS-Steering and LoRA adapter. Training $Sr$ and $Sd$ jointly collapses COS-Steering into SFT with LoRA. SFT struggles to acquire abstract notions such as safety or truthfulness; this limitation has been confirmed by baselines in our paper[1][2].
>
> [1] Kenneth Li, Oam Patel, Fernanda B. Viégas, Hanspeter Pfister, Martin Wattenberg: Inference-Time Intervention: Eliciting Truthful Answers from a Language Model. NeurIPS 2023
>
> [2] Zhangchen Xu, Fengqing Jiang, Luyao Niu, Jinyuan Jia, Bill Yuchen Lin, Radha Poovendran: SafeDecoding: Defending against Jailbreak Attacks via Safety-Aware Decoding. ACL 2024
>
> ---
>
> We are very grateful for this detailed list of suggestions, which will significantly improve the paper's quality and readability. If any concerns remain, or if new questions arise, we would be grateful to hear from you.

---

> > ### Comment · Reviewer_kKNM · 2025-11-25
> >
> > Thank you very much for your detailed responses. My first two concerns have been addressed, but the third concern remains.
> >
> > The plots show that $S_r$ is selective on harmful versus benign inputs, but this alone does not show that the basis vectors in $S_d$ themselves encode safety-relevant directions. Since $S_r$ is trained to separate positive and negative examples, it could in principle learn to use almost any fixed basis $\{S_d\}$, even if those directions were random or unrelated to toxicity. In that case, COS-Steering would behave like a low-rank safety adapter, and the selectivity of $S_r$ would not prove that $S_d$ forms a genuine safe-steering subspace.
> >
> > Your motivation about decoupling the selector $S_r$ from the representation $S_d$ and avoiding LoRA-style blending is reasonable, but I still do not see empirical evidence that $S_d$ is better than directions from a standard LoRA adapter or a generic low-rank subspace.
> >
> > To make this point more convincing, I would find one of the following very helpful:
> >
> > - Replace $S_d$ with a random or clearly non-safety basis of the same rank, retrain $S_r$ with the same budget, and show that ASR / ORR / benign quality become clearly worse.
> > - Show, for each $S_d$ vector, that the inputs in its “top-10% activation range” are strongly enriched for specific harmful categories or attack types, and that this enrichment is stronger than for a random basis. It would also help to state explicitly in the paper that “top-10% activation range” means: for each feature, take the 10% of inputs with highest weight and examine the label distribution in that subset.
> > - Train a simple linear probe on $S_d$ activations alone to predict harmful vs benign, and compare against probes on raw hidden states or random bases. If probes on $S_d$ perform better, this would support the claim that COS-Steering has distilled a meaningful safety subspace.
> >
> > At the moment, COS-Steering still feels quite close to a decoupled LoRA with extra structure. For me, the method would be a stronger contribution if the final version could clearly show either (i) better interpretability of the learned subspace, or (ii) clearly stronger steering performance than competitive LoRA / SFT baselines with similar capacity.

---

> > > ### Author Response · Authors · 2025-11-26
> > >
> > > Thank you for your further discussion and feedback on our work!
> > > We will demonstrate why $Sd$ constitutes an **interpretable** safety subspace from both theoretical analysis and experimental evidence.
> > >
> > > **Theoretical Analysis**
> > >
> > > $Sd$ extracts the safe directions from steering vectors that are already capable of safety steering in specific contexts. After several epochs of training, the vectors obtained in Step 1 can steer the model to produce harmless responses for the majority of bad cases, as shown in Figure 3. These extracted vectors encode salient safety-steering signals within the respective contexts. By extracting the principal components of these vectors, $Sd$ naturally forms a rich safety-steering subspace.
> > >
> > > **Experimental Evidence**
> > >
> > > **Probing.** To further validate the effectiveness of $Sd$, we trained a simple linear probe on the raw hidden states, LoRA activation and the $Sd$ activations to distinguish between harmless and harmful scenarios. The results are presented below.
> > >
> > > | Layer | Metric     |Raw Hidden States|LoRA Activation|$Sd$ Activation|Δ      |
> > > |-------|------------|-----------------|---------------|---------------|-------|
> > > | 16    | Accuracy   | 88.7%           | 89.4%         | 94.8%         | +5.4% |
> > > | 16    | Precision  | 91.5%           | 91.3%         | 95.3%         | +3.8% |
> > > | 16    | Recall     | 88.5%           | 87.4%         | 95.8%         | +7.3% |
> > > | 16    | $F_1$ Score| 89.3%           | 87.6%         | 95.5%         | +6.2% |
> > > | 20    | Accuracy   | 83.0%           | 80.5%         | 91.2%         | +8.2% |
> > > | 20    | Precision  | 90.3%           | 82.5%         | 92.0%         | +1.7% |
> > > | 20    | Recall     | 81.3%           | 77.4%         | 92.9%         | +11.6%|
> > > | 20    | $F_1$ Score| 81.6%           | 75.4%         | 92.2%         | +10.6%|
> > > | 24    | Accuracy   | 82.0%           | 89.4%         | 95.9%         | +6.5% |
> > > | 24    | Precision  | 89.7%           | 91.6%         | 96.7%         | +5.1% |
> > > | 24    | Recall     | 80.3%           | 87.0%         | 96.5%         | +9.5% |
> > > | 24    | $F_1$ Score| 80.4%           | 86.6%         | 96.5%         | +9.9% |
> > >
> > > The results show that $Sd$ activations are more sensitive to input harmfulness than the raw hidden states and LoRA activations. And this property of $Sd$ activations is achieved by encoding safety-relevant directions explicitly.
> > >
> > > **The Uninterpretability of LoRA.** Following the setup in Figure 5 and Section 4.5, we examined the intermediate outputs of a LoRA adapter that shares the same architecture as COS-Steering; results are shown in Figure 14 of Appendix C.4. Unlike COS-Steering, the LoRA features exhibit no discernible specialization toward any particular attack method—most activations follow similar distributions. In contrast, $Sd$—serving as a dedicated safety subspace—offers greater interpretability than the LoRA adapter.
> > >
> > > We are very grateful for your detailed review, which will significantly improve the paper's quality and readability. The discussion above has been added to Appendix C.4 and is highlighted in blue. If any concerns remain, or if new questions arise, we would be grateful to hear from you.

---

> > > > ### Comment · Reviewer_kKNM · 2025-11-27
> > > >
> > > > Thank you for your response. The additional experimental evidence is very promising. I have therefore increased my score.

---

> > > > > ### Author Response · Authors · 2025-11-27
> > > > >
> > > > > Dear reviewer kKNM:
> > > > >
> > > > > Thank you very much for your detailed comments; we believe the paper is much improved by their addition, and we very much appreciate your generous improvement in score.

---

### Official Review · Reviewer_UDHN · 2025-10-29

**Soundness:** 3
**Presentation:** 3
**Contribution:** 3
**Rating:** 8
**Confidence:** 3

**Summary:**

This paper proposes COS-Steering, a dynamic method that allows inputs to determine their own safety-steering direction within the semantic space. Experiments show it maintains strong refusal of harmful prompts while causing minimal side effects on benign queries.

**Strengths:**

Solid Theory and Motivations
- The paper clearly states the weakness of the previous method.
- provides solid observations of `Activation shifting driven by jailbreak attack` in the vision of LLM internal activation state.

Comprehensive Evaluations and In-depth Analysis
- The paper not only proves the effectiveness of the proposed method, but also provides non-trivial insights in its working mechanism.

Good Demonstrations and writing

**Weaknesses:**

There are minor weaknesses. Figure 2 should have a clearer resolution. Line 425 has a reference link failure.

**Questions:**

As the experiments only include the Llama2 series and the Vicuna series, why the latest version (at least one latest series) are not included, e.g., Llama3 series and Qwen3?

---

> ### Author Response · Authors · 2025-11-18
> **Rebuttal for Reviewer UDHN**
>
> ### **Rebuttal for Reviewer UDHN**
>
> We sincerely thank you for insightful feedback and constructive suggestions. These comments are invaluable for enhancing the clarity, rigor, and impact of our work. We have carefully considered each point and will address them below.
>
> ---
>
> ### **Response to Weakness: Syntax Errors**
>
> > **Reviewer's Concern:** There are minor weaknesses. Figure 2 should have a clearer resolution. Line 425 has a reference link failure.
>
> Figure 2 has been replaced with a high-resolution version, and the broken link has been fixed.
>
> ---
>
> ### **Response to Question: Experiments on Llama-3-8B**
>
> > **Reviewer's Concern:** As the experiments only include the Llama2 series and the Vicuna series, why the latest version (at least one latest series) are not included, e.g., Llama3 series and Qwen3?
>
> Experiments are conducted on Llama-3-8B; results are reported below.
>
> | Method  | Advbench | HarmfulQA|
> |:-------:|:--------:|:-------:|
> |Llama3-8B|  0.10    |   0.05  |
> |+COS-Steering|  0.01  |  0.01 |
>
> ---
>
> We are very grateful for this detailed list of suggestions, which will significantly improve the paper's quality and readability. If any concerns remain, or if new questions arise, we would be grateful to hear from you.

---

### Official Review · Reviewer_JUgT · 2025-11-01

**Soundness:** 2
**Presentation:** 2
**Contribution:** 3
**Rating:** 2
**Confidence:** 4

**Summary:**

The paper proposes COS-Steering, a context-specific activation steering method that learns a safe-steering subspace and applies input-conditioned linear combinations of basis vectors to steer an LLM’s activations only when needed. Concretely: (1) it trains multiple steering vectors on small shards of harmful prompts to elicit a harmless prefix; (2) compresses these steering vectors with a sparse autoencoder (SAE) to obtain a compact set of basis vectors; and (3) learns a tiny Steering Representor that maps current hidden states to weights over those bases, with an Orthogonal Representation Penalty.

**Strengths:**

1. The method moves beyond static steering by conditioning steering directions on input context, addressing attack diversity.
2. The design explicitly creates distribution shift by mixing attacks across datasets.

**Weaknesses:**

1. Steering vectors are trained to produce a specific harmless prefix. While the paper discusses why this can flip harmfulness effectively, it’s still a narrow textual proxy for safety and may overfit to that phrasing/distribution; consequences for content-level safety beyond prefix control deserve deeper auditing.

2. The SAE is used for compression rather than monosemantic discovery; however, the semantics of the learned bases are only lightly analyzed. More direct probes would strengthen interpretability claims.

3. We see hyper-parameter settings and a note that 2 epochs are “prudent,” but key ablations are missing: (i) How sensitive are results to the SAE bottleneck size? (ii) What if we remove POR or vary its strength? (iii) How many steering vectors/shards are needed before diminishing returns? (iv) Layer placement sensitivity beyond 9–30?

4. Reliance on Llama-Guard2 and GPT-judge templates can create evaluation circularity. Cross-checking with alternative safety classifiers would reduce metric bias.

5. Experiments are conducted only on Vicuna-7B and LLaMA-2-7B, which are now relatively dated models. Given the rapid evolution of open-weight LLMs, results may not generalize to newer architectures with different safety and activation structures.

6. The motivation of claiming that static, category-based steering fails under adversarial prompt variations is insufficiently substantiated. The paper does not clearly demonstrate why or how these category boundaries collapse under jailbreak transformations. More in-depth analysis or explanation of activation drift would make the premise more convincing.

7. Only BERTScore and KL metrics are reported. Broader benign performance (e.g., instruction-following, creativity) should be measured.

**Questions:**

1. How does COS-Steering perform against paraphrased or long-context adversarial prompts that delay harmful intent?

2. What is the minimal number of steering vectors or shards required before performance saturates?

---

> ### Author Response · Authors · 2025-11-18
> **Rebuttal for Reviewer JUgT 1/4**
>
> ### **Rebuttal for Reviewer JUgT**
>
> We sincerely thank you for insightful feedback and constructive suggestions. These comments are invaluable for enhancing the clarity, rigor, and impact of our work. We have carefully considered each point and will address them below.
>
> ---
>
> ### **Response to W1: Effectiveness of Optimizing Harmless Prefix**
>
> > **Reviewer's Concern:** Steering vectors are trained to produce a specific harmless prefix. While the paper discusses why this can flip harmfulness effectively, it's still a narrow textual proxy for safety and may overfit to that phrasing/distribution; consequences for content-level safety beyond prefix control deserve deeper auditing.
>
> We thank the reviewer for this critical question. We acknowledge this critical issue for COS-Steering's effectiveness, which we will address theoretically and experimentally.
>
> **Theoretical Aspect**
>
> Harmless prefixes is not mere **surface-form memorization**; instead, it activates the model's **latent safety awareness** under harmful contexts.
> The prefixes we select are the most frequent ones appearing in the model's safe responses to harmful queries.
> And these prefixes are the most frequent safe openers, because they are repeatedly paired with harm-averse responses throughout pre-training and alignment.
> When COS-Steering trains a vector to maximize the likelihood of these prefixes, it learns to shift the model's hidden state into the region of high safety preference under harmful contexts.
> At **every token step**, COS-Steering keeps the hidden state in view, and applies a safety steering whenever necessary.
>
>
> **Experimental Aspect**
>
> Experiments show that our method sustains safety well beyond the prefix-controlled span.
> (1) In the experiments of Table 1, all evaluations of harmfulness are conducted at the **context-level** with the complete response (up to the maximum length).
> (2) COS-Steering demonstrates robust resistance against attacks that employ **delayed harmful intent**. In Table 2, DRA[1]-a strong jailbreak that initially disguises malicious instructions as an innocuous anagram-requires the intermediate chain "extract characters → reconstruct → emit" before its harmful intent surfaces. Llama-2-7B, endowed with strong safety preferences, detects a subset of these latent intents; COS-Steering further amplifies this detection capability. An analogous enhancement is observed on Vicuna-13B, whose native safety preference is weaker.
>
> Nevertheless, your valuable feedback has alerted us to the necessity of clarifying this issue; we have accordingly added an explicit discussion in the revised paper. The corresponding exposition has been added to Appendix A.5 and is highlighted in blue.
>
> [1] Tong Liu, Yingjie Zhang, Zhe Zhao, Yinpeng Dong, Guozhu Meng, Kai Chen: Making Them Ask and Answer: Jailbreaking Large Language Models in Few Queries via Disguise and Reconstruction. USENIX Security Symposium 2024
>
> ---
>
> ### **Response to W2: Semantic Observation of SAE**
>
> > **Reviewer's Concern:** The SAE is used for compression rather than monosemantic discovery; however, the semantics of the learned bases are only lightly analyzed. More direct probes would strengthen interpretability claims.
>
> Figure 5 in the paper already visualizes the **minimal interpretable semantic units** attainable under **the present SAE-dimensional setting**.
> As you noted, the SAE is used for compression rather than monosemantic discovery; this implies that the SAE encodes macro-level semantics-or, as we prefer to term them, foundational safety strategies. Although we attempted to discover finer-grained semantics, there is no interpretation observed beyond the category level under these SAE-dimensional setting.

---

> ### Author Response · Authors · 2025-11-18
> **Rebuttal for Reviewer JUgT 2/4**
>
> ### **Response to W3: Ablations**
>
> > **Reviewer's Concern:** We see hyper-parameter settings and a note that 2 epochs are "prudent", but key ablations are missing: (i) How sensitive are results to the SAE bottleneck size? (ii) What if we remove  $P_{OR}$    or vary its strength? (iii) How many steering vectors/shards are needed before diminishing returns? (iv) Layer placement sensitivity beyond 9-30?
>
> (i) & (iii) Figure 9 is added to report the impact on both reconstruction loss and ASR. The purpose of tuning SAE width is to suppress reconstruction error and redundant features. Since the SAE is trained offline without invoking the LLM, the cost is negligible-under one minute on a single CPU. The dimension can be set to the smallest value at which the reconstruction loss plateaus after a few exploratory runs.
>
> (ii) Figures 7 and 10 are added to contrast $Sd$ feature activations with and without  $P_{OR}$   . Without  $P_{OR}$   , many $Sd$ features remain strongly activated even in benign contexts.
>
> (iv) The principle is to exclude every layer that processes low-level features. In LLMs, the lower layers attend primarily to syntax and other superficial cues, which we leave untouched. Layer 9 is therefore adopted as the lowest intervened layer, following previous partial-layer steering methods[2].
>
> The corresponding exposition has been added to Appendix A.4 and is highlighted in blue.
>
> [2] Zhihao Xu, Ruixuan Huang, Changyu Chen, Xiting Wang: Uncovering Safety Risks of Large Language Models through Concept Activation Vector. NeurIPS 2024
>
> ### **Response to W4: Cross-checking with alternative safety classifiers**
>
> > **Reviewer's Concern:** Reliance on Llama-Guard2 and GPT-judge templates can create evaluation circularity. Cross-checking with alternative safety classifiers would reduce metric bias.
>
> We employed ShieldLM-14B-Qwen as the surrogate classifier. Evaluation results are as follows:
>
> | Type               | Method         | Advbench ASR↓ Llama Guard | Advbench ASR↓ Shield LM |  HarmfulQA ASR↓ Llama Guard | HarmfulQA ASR↓ Shield LM |
> |--------------------|----------------|---------------------------|-------------------------|-----------------------------|--------------------------|
> | Decoding Time      | Self-Reminder  | 0.40                      | 0.44                    |  **0.23**                   | **0.33**                 |
> | Decoding Time      | SafeDecoding   | 0.44                      | 0.69                    |  0.46                       | 0.73                     |
> | Activation Eng.    | MeanCentring   | 0.61                      | 0.73                    |  0.46                       | 0.47                     |
> | Activation Eng.    | BiPO           | **0.32**                  | **0.31**                |  0.34                       | 0.37                     |
> | Activation Eng.    | ITI            | 0.39                      | 0.57                    |  0.28                       | 0.39                     |
> | Activation Eng.    | CAST           | 0.76                      | 0.85                    |  0.52                       | 0.78                     |
> | **Ours**           | **COS-Steering** | **0.08**                | **0.12**                |  **0.05**                   | **0.13**                 |
>
> **Evaluation Circularity.** Using an LLM from the same family as the judge to score its own behavior risks in-group bias: shared data, style, and objective functions encourage the judge to favor its siblings, yielding artificially low ASR in our setting. To mitigate this bias, we re-compute ASR with ShieldLM-14B-Qwen from the Qwen family serving as an external judge; the results are reported in the Table 4. Most methods exhibit a noticeable ASR increase, yet COS-Steering's rise is modest and still outperforms all baselines in ASR.
>
> These results are also reported in Table 4 of the paper.
>
> ---
>
> ### **Response to W5: Experiments on Llama-3-8B**
>
> > **Reviewer's Concern:** Experiments are conducted only on Vicuna-7B and LLaMA-2-7B, which are now relatively dated models. Given the rapid evolution of open-weight LLMs, results may not generalize to newer architectures with different safety and activation structures.
>
> Experiments are conducted on Llama-3-8B; results are reported below.
>
> | Method  | Advbench | HarmfulQA|
> |:-------:|:--------:|:-------:|
> |Llama3-8B|  0.10    |   0.05  |
> |+COS-Steering|  0.01  |  0.01 |

---

> ### Author Response · Authors · 2025-11-18
> **Rebuttal for Reviewer JUgT 3/4**
>
> ### **Response to W6: The Cause of Activation Shift under Jailbreak Attacks**
>
> > **Reviewer's Concern:** The motivation of claiming that static, category-based steering fails under adversarial prompt variations is insufficiently substantiated. The paper does not clearly demonstrate why or how these category boundaries collapse under jailbreak transformations. More in-depth analysis or explanation of activation drift would make the premise more convincing.
>
> Adversarial prompt variations are more discernible than shifts across harm categories. Each attack drastically rewrites the original prompt-embedding jailbreak templates, injecting bespoke logic, or inserting rare textual artifacts-whereas harm categories merely switch topical labels. Because hidden states encode the model's internal context, they inevitably register these coarse alterations. The resulting textual structures, logical schemes, and stylistic quirks are captured as distinctive activation signatures, driving apart the hidden states of prompts that share the same harm category but differ in adversarial form.
>
> The corresponding exposition has been added to Appendix A.1 and is highlighted in blue.
>
> ---
>
> ### **Response to W7: Evaluation Metrics**
>
> > **Reviewer's Concern:** Only BERTScore and KL metrics are reported. Broader benign performance (e.g., instruction-following, creativity) should be measured.
>
> Using KL divergence to quantify the shift in the model's output distribution by comparing logits before and after alignment is an established practice in the literature. To comprehensively evaluation, perplexity and over-refusal rates have been incorporated into the evaluation and are reported in the table above and Table 4 in paper.
>
> | Type               | Method         | Advbench PPL↓ | Advbench ORR↓ | HarmfulQA PPL↓ | HarmfulQA ORR↓ |
> |--------------------|----------------|---------------|---------------|----------------|----------------|
> | Decoding Time      | Self-Reminder  | 15.49         | **0.01**      | 5.84           | **0.01**       |
> | Decoding Time      | SafeDecoding   | 7.18          | **0.02**      | **1.07**       | 0.04           |
> | Activation Eng.    | MeanCentring   | **7.14**      | 0.21          | 25.25          | 0.09           |
> | Activation Eng.    | BiPO           | 45.49         | 0.05          | 24.17          | **0.01**       |
> | Activation Eng.    | ITI            | 13.70         | **0.01**      | **4.81**       | 0.20           |
> | Activation Eng.    | CAST           | 32.67         | **0.02**      | 32.67          | **0.02**       |
> | **Ours**           |**COS-Steering**| **3.83**      | **0.02**      | 7.99           | **0.01**       |
>
> **Evaluation of generation quality.** Table 1 uses KL divergence and BERTScore to quantify the deviation from the original model's distribution after alignment. As a complementary metric, perplexity is adopted to directly assess generation quality on benign prompts, shown in Table above. The results show that COS-Steering keeps perplexity at a low level overall. To avoid evaluation circularity, the scores are computed with Qwen3-8B.
>
> **Over-refusal rate.** Over-refusal rate is a key metric for gauging the excessive sensitivity of a LLM's safety mechanism; it measures the proportion of benign prompts that the model incorrectly refuses to answer. Table above shows that COS-Steering maintains a low over-refusal rate.

---

> ### Author Response · Authors · 2025-11-18
> **Rebuttal for Reviewer JUgT 4/4**
>
> ### **Response to Q1: Against Delay Harmful Intent**
>
> > **Reviewer's Concern:** How does COS-Steering perform against paraphrased or long-context adversarial prompts that delay harmful intent?
>
> This part is addressed in W1:
>
> COS-Steering demonstrates robust resistance against attacks that employ **delayed harmful intent**. In Table 2, the DRA[1] attack-a strong jailbreak that initially disguises malicious instructions as an innocuous anagram-requires the intermediate chain "extract characters → reconstruct → emit" before its harmful intent surfaces. Llama-2-7B, endowed with strong safety preferences, detects a subset of these latent intents; COS-Steering further amplifies this detection capability. An analogous enhancement is observed on Vicuna-13B, whose native safety preference is weaker.
>
> [1] Tong Liu, Yingjie Zhang, Zhe Zhao, Yinpeng Dong, Guozhu Meng, Kai Chen: Making Them Ask and Answer: Jailbreaking Large Language Models in Few Queries via Disguise and Reconstruction. USENIX Security Symposium 2024
>
> ---
>
> ### **Response to Q2: Dimension of SAE**
>
> > **Reviewer's Concern:** What is the minimal number of steering vectors or shards required before performance saturates?
>
> This part is addressed in W3:
>
> Figure 9 is added to report the impact on both reconstruction loss and ASR. The purpose of tuning SAE width is to suppress reconstruction error and redundant features. Since the SAE is trained offline without invoking the LLM, the cost is negligible-under one minute on a single CPU. The dimension can be set to the smallest value at which the reconstruction loss plateaus after a few exploratory runs.
>
> ---
>
> We are very grateful for this detailed list of suggestions, which will significantly improve the paper's quality and readability. If any concerns remain, or if new questions arise, we would be grateful to hear from you.

---

> > ### Comment · Reviewer_JUgT · 2025-11-18
> >
> > Thank the authors for their detailed response. I think it addressed most of my concerns, so I have increased my rating.

---

> > > ### Author Response · Authors · 2025-11-19
> > >
> > > Dear reviewer JUgT:
> > >
> > > Thank you very much for your detailed comments; we believe the paper is much improved by their addition, and we very much appreciate your generous improvement in score.

---

### Author Response · Authors · 2025-11-29
**Summary of Rebuttal (8422 → 8466) (1/2)**

Dear AC, thank you for taking the time to read the reviews. The API security incident has created a lot of trouble for our submission—one that saw a sharp score increase during rebuttal (8422 → 8466)—and has also added extra work for you. To help you quickly understand the reviewers’ main concerns and what happened during rebuttal, we provide a very brief summary of the reviewers' concerns, a timeline of discussion. We hope this helps reduce your workload, and also hope that our work and the efforts made by the reviewers during the rebuttal period will be evaluated fairly and objectively.

Too long; didn’t read: summary of reviewer concerns, the timeline of discussion.

### **Summary of Reviewer Concerns**

To reduce your workload, we provide a concise summary of the reviewers’ concerns and our responses; reading it first may help navigate the rebuttal efficiently.
Most reviewers have **a validity-threatening major concern** plus **other concerns**. W, Q, R denote weakness, question, and response, respectively.

**Reviewer JUgT (Score: 2 → 6)**

*Major Concern.*

W1 & Q1: Learning specific harmless prefix is narrow proxy for safety, consequences for inability when facing delay harmful intent.
R: We theoretically and empirically show that learning a narrow set of harmless prefixes awakens the model’s intrinsic safety mechanism, conferring resistance against delayed-intent attacks.

*Other Concerns.*

W2: Finer-grained interpretability.
R: Already the finest possible.

W3: More ablations.
R: Added in Appendix A.4.

W4: Evaluation circularity.
R: Added in Table 4.

W5: More recently released model.
R: Added in rebuttal.

W6: Explanation of the motivating phenomenon.
R: Added in Appendix A.1.

W7: More metrics to evaluate the impact on benign scenarios.
R: Added in Table 4.

Q1/Q2: Follow-ups on W1/W3.
R: Answered together with W1/W3.

**Reviewer UDHN (Score: 8)**

W: Syntax errors.
R: Fixed.

Q: More recently released model.
R: Added in rebuttal.

**Reviewer kKNM (Score: 2 → 6)**

*Major Concern: Evidence of the difference between COS-Steering and LoRA adapters*

W3: Lack of clear evidence on $Sr$ choosing $Sd$ decoder vectors selectively.
R: Figure 5 demonstrates selective-activation evidence across various attack methods, while supplementary Figure 7 provides analogous evidence for benign-scene exclusion.

Q3: Difference between COS-Steering and a LoRA adapter.
R: The baselines in our paper have already shown that SFT with LoRA has limited capacity to capture comprehensive safety-related features.

Further dicussion: Evidence is required that COS-Steering offers superior interpretability or steering efficacy compared to LoRA.
R: We show COS-Steering is more interpretable than LoRA, both in theory and in experiment.

*Other Concerns.*

W1: Missing LLM usage and reproducibility statements.
R: Added in Section 6.

W2: More metrics to evaluate the impact on benign scenarios.
R: Added in Table 4.

W4: Syntax errors.
R: Fixed.

Q1 & Q2: Details of Figure 5.
R: Added caption + inline explanation.

**Reviewer Mobr (Score: 4)**

*Major Concern*: PaCE (NeurIPS 2024) challenges our static assumption claim, clouding the paper’s contribution.
R: PaCE does not challenge our static assumption claim. Actually, it **suffers** from the core flaw of the static premise: **over-reliance on hand-crafted semantic categories**.

*Other Concerns.*

W1 & W2: Details of method.
R: Explain directly.

W3: Usage of metrics.
R: Our evaluation aligns exactly with the reviewer’s suggestion.

W4: More recently released model.
R: Added in rebuttal.

W5: Missing of statistical uncertainty.
R: Added in Table2.

W6: Syntax errors.
R: Fixed.

---

> ### Author Response · Authors · 2025-11-29
> **Summary of Rebuttal (8422 → 8466) (2/2)**
>
> ### **Timeline of Discussion**
>
> Reviewer JUgT and reviewer kKNM both raised their scores from 2 to 6; the remaining reviewers did not respond before the freeze. Below are the discussion and timeline (all times in EST as shown in OpenReview's report).
>
> 07:27 a.m. EST, 12 Nov 2025 – Discussion opens.
>
> ↓
>
> 02:07 a.m. EST, 18 Nov 2025 – Full rebuttal + revised PDF submitted.
>
> ↓
>
> 12:46 p.m. EST, 18 Nov 2025 – Reviewer JUgT confirms issues resolved; score raised (2 → 6).
>
> ↓
>
> 04:05 a.m. EST, 25 Nov 2025 – Reviewer kKNM: minor concerns resolved, but asks for explicit evidence that COS-Steering is more interpretable / steerable than LoRA.
>
> ↓
>
> 12:11 a.m. EST, 26 Nov 2025 – We reply with theoretical + experimental evidence of superior interpretability.
>
> ↓
>
> 02:35 a.m. EST, 27 Nov 2025 – Reviewer kKNM confirms issues resolved; score raised (2 → 6).
>
> ↓
>
> 10:09 a.m. EST, 27 Nov 2025 – API vulnerability reported & widely exploited.
>
> ↓
>
> 11:10 a.m. EST, 27 Nov 2025 – Program Chairs and Workflow Chair notified of resolution.
>
> Both reviewers raised their scores well before the large-scale API-security incident: JUgT replied almost nine days earlier, and kKNM’s first revision already signaled a willingness to increase the score.
> We apologize that non-academic content has become part of this rebuttal, yet we feel compelled to defence for the time and effort invested by both the reviewers and ourselves.
> An anonymized screenshot of the pre-incident scores is provided in the Figure 16 for reference temporarily.
>
> Dear AC, thank you very much for reading the reviews. We sincerely ask you to consider whether our rebuttal has fully resolved the reviewers’ concerns and thus warrants the significant score increase seen during rebuttal (8422 → 8466). If any concerns remain, or if new questions arise, we would be grateful to hear from you. Finally, we once again thank all the reviewers for the invaluable efforts; your constructive feedbacks are essential for improving the rigor, clarity, and impact of our work.

---

### Meta-Review · Area_Chair_qvdN · 2026-01-06

**Summary:**

This paper proposes using learned steering vectors in an input-specific manner via an attention mechanism.

In the paper, there is a reference that is hallucinated:
Francesco Croce, Matthew Finzi, Stefano Ermon, and Nicholas Carlini. Advbench: A benchmark for adversarial robustness. arXiv preprint arXiv:2107.04457, 2021.

I assume the authors wanted to cite:
Zou, Andy, et al. "Universal and transferable adversarial attacks on aligned language models." arXiv preprint arXiv:2307.15043 (2023).

This hallucinated reference is very problematic since LLM-based search engines now can use (and do use it, I checked myself) that ICLR submission (Submission3381) as a link to justify the existence of (Francesco Croce, Matthew Finzi, Stefano Ermon, and Nicholas Carlini. Advbench: A benchmark for adversarial robustness) reference when searching for Advbench.

The reviewer mentioned a significant lack of evaluation of the utility and compliance (non over-refusal) of the model after training. I completely agree that it is a significant weakness of the work since of model that is always refusing is 100% safe (but useless).

I believe these concerns are still outstanding because of the lack of details about the over-refusal evaluation proposed in the rebuttal and the lack of evaluation on other datasets beyond the ones used for training.

**Reviewer Concerns:**

The reviewer's concern, raised by JUgT, kKNM and Morb regarding the lack of evaluation of benign queries, has not been satisfactorily addressed in my opinion.

> Table 4 : OOR denotes the over-refusal rate on harmless queries.

In Table 4 (and the rest of the paper), there are no details about the harmless dataset used for the queries and the paper does not evaluate capabilites (KL on Harmbench is not sufficient as an evaluation). The paper mentions Advbench and HarmfulQA, which do not contain harmless queries (e.g. how to kill a Python process). Moreover, the evaluation of capabilities (and overrefusal) should be significantly more thorough and conducted on other datasets than the refusal ones. For instance, for capabilities, Zou et al. (circuit breaker) consider MT-Bench and Open LLM, while Bhardwaj and Poria (2023) consider TruthfulQA, MMLU, and BBH. For measuring over refusal (they call it compliance) Sheshadri et al. (2024) use the Ultrachat dataset (https://arxiv.org/pdf/2407.15549)

**Reviewer Scores:**

If I had been able to discuss this issue of utility and over refusal evalutation with the reviewers (JUgT, kKNM and Morb), I believed they would have kept their score.

---

### Decision · Program_Chairs · 2026-01-26

Reject